# CNVs Associated with Different Clinical Phenotypes of Psoriasis and Anti-TNF-Induced Palmoplantar Pustulosis

**DOI:** 10.3390/jpm12091452

**Published:** 2022-09-04

**Authors:** Alejandra Reolid, Antonio Sahuquillo-Torralba, Ancor Sanz-García, Rafael Botella-Estrada, Ester Muñoz-Aceituno, Mar Llamas-Velasco, Jorge García-Martínez, Esteban Daudén, Francisco Abad-Santos, María C. Ovejero-Benito

**Affiliations:** 1Department of Dermatology, Instituto de Investigación Sanitaria La Princesa (IIS-IP), Hospital Universitario de la Princesa, 28006 Madrid, Spain; 2Department of Dermatology, Instituto de Investigación Sanitaria La Fe (IIS La Fe), Hospital Universitario y Politécnico La Fe, 46026 Valencia, Spain; 3Data Analysis Unit, Instituto de Investigación Sanitaria La Princesa, Hospital Universitario de la Princesa, (IIS-IP), 28006 Madrid, Spain; 4Facultad de Medicina, Universidad de Valencia, 46010 Valencia, Spain; 5Clinical Pharmacology Department, Hospital Universitario de la Princesa, Instituto de Investigación Sanitaria la Princesa (IIS-IP), 28006 Madrid, Spain; 6Instituto Teófilo Hernando, 28029 Madrid, Spain; 7Faculty of Medicine, Universidad Autónoma de Madrid (UAM), 28029 Madrid, Spain; 8Centro de Investigación Biomédica en Red de Enfermedades Hepáticas y Digestivas (CIBERehd), Instituto de Salud Carlos III, 28222 Madrid, Spain

**Keywords:** pharmacogenomics, psoriasis genetics methylation array, biological drugs, adverse effect prediction

## Abstract

Background: Psoriasis can present different phenotypes and could affect diverse body areas. In contrast to the high effectiveness of biological drugs in the treatment of trunk and extremities plaque psoriasis, in palmoplantar phenotypes and in plaque scalp psoriasis, these same drugs usually have reduced efficacy. Anti-TNF drugs could induce the appearance of palmoplantar pustulosis (PPP) in patients with other inflammatory diseases. The objective of this study is to identify if there are DNA Copy Number Variations (CNVs) associated with these different clinical phenotypes, which could justify the differences found in clinical practice. Moreover, we intend to elucidate if anti-TNF-induced PPP has a similar genetic background to idiopathic PPP. Methods: Skin samples were collected from 39 patients with different patterns of psoriasis and six patients with anti-TNF-induced PPP. The CNVs were obtained from methylation array data (Illumina Infinium Human Methylation) using the conumee R package. Results: No significant CNVs were found between the different phenotypes and the locations of psoriasis compared. Nevertheless, we found two significant bins harboring five different genes associated with anti-TNF-induced PPP in patients with a different background other than psoriasis. Conclusions: Our results may help to predict which patients could develop anti-TNF-induced PPP.

## 1. Introduction

Psoriasis is the most common dermatological disease in adults, with a prevalence of approximately 3% worldwide [1]. The term psoriasis encompasses different clinical patterns of disease: guttate, erythrodermic, inverse, pustular, palmoplantar, and plaque psoriasis, with the last form of the disease being the most frequent [2]. However, sometimes it can be limited exclusively to the scalp, palms, and soles with or without the presence of the typical psoriasis Vulgaris plaques. The palmoplantar forms can manifest as erythematous-scaly-hyperkeratotic plaques (hyperkeratotic palmoplantar psoriasis) or as sterile pustules (palmoplantar pustular psoriasis-PPP). In contrast to the high effectiveness of new biological drugs in the treatment of psoriasis Vulgaris of the trunk and extremities, in both palmoplantar phenotypes and in the lesions located predominantly on the scalp, these drugs usually have a slow and limited or null effect [3,4]. The marked clinical differences and therapeutic response to the same treatments between these subgroups suggest that psoriasis should be considered as a spectrum of diseases rather than as a single entity [5]. Thus, it is important to unveil potential genetic factors that determine the location and phenotype of psoriasis lesions.

Psoriasis is currently considered a systemic inflammatory disease with a multifactorial polygenic inheritance [6]. Genetic susceptibility is determined primarily by 65 loci known as psoriasis susceptibility regions (PSORS) that have been linked to psoriasis risk [7]. PSORS1, with the target allele HLA-C:06:02, contributes to 30–50% of genetic susceptibility, presenting its carriers with a 3–4 times higher risk of suffering from the disease [8]. In pustular forms, mutations and polymorphisms have been described in the genes CARD14, AP1S3, and IL36RN [9,10,11], although the proportion of subjects harboring IL36RN-mutated alleles seems to be higher in generalized pustular psoriasis (GPP) than in palmoplantar pustulosis (PPP), the prevalence of AP1S3 mutations seems to be similar across these disease types and CARD14 mutations are observed in only a small minority of cases [11,12,13].

However, not all patients carrying HLA-C:06:02 or the above-mentioned genes suffer from the disease [8]. This fact, together with the absence of total concordance between monozygotic twins, suggests that, although the influence of these genetic factors in psoriasis is evident, there must be other genetic or epigenetic determinants of the disease and its phenotypic variants.

Each autosomal gene has two different copies originating from each parent. However, certain DNA regions have Copy Number Variations (CNVs) caused by duplications or deletions that result in a higher or lower number of copies [14,15]. Those CNVs can modify the genome structure and transposable elements within a region, thus affecting gene expression. They have been reported in healthy individuals [16,17] and in different pathologies such as cancer or autoimmune diseases [14] they can also be used as diagnostic biomarkers in several diseases [16,17]. Finally, CNVs seem to be involved in psoriasis pathogenesis through the modification of the expression of keratinocyte proteins involved in their cellular integrity and intercellular adhesion or its innate defense capacity, which would result in keratinocytes predisposed to induce psoriasis development [18,19,20,21,22]. The homozygous or heterozygous deletion of the late cornified envelope genes LCE3B and LCE3C is associated with an increase in the risk of psoriasis [18,23]. An increment of ≥3 copies in the CNVs of CCL4/CCL4L is associated with psoriasis severity, whereas moderate disease correlated with a lower CNV (≤2 copies); specifically, the CCL4L1 allele frequency is higher in severe psoriasis, whereas CCL4L2 is more frequent in patients with milder disease. CCL4/CCL4L genes encode for chemokines that stimulate T helper type 1 (Th1) cells, T regulatory (Treg) cells, monocytes, and dendritic cells (DC) [22]. On the other hand, the presence of certain CVNs has been associated with a better response to certain biological therapies, mainly anti-TNF [24,25]. However, these studies have focused on the discovery of characteristic CNVs of psoriasis Vulgaris using blood samples and not the tissue that expresses the disease, which on many occasions presents in a focal manner on the elbows and knees or with the exclusive involvement of the scalp or palmoplantar area. Therefore, we consider that it might be more relevant to study whether there are common or differential CNVs between the different clinical phenotypes of psoriasis in the tissue that clinically expresses the disease.

Apart from the above-related clinical types of psoriasis, anti-TNF drugs can induce the development of psoriasiform reactions [26,27], of which PPP is the most frequent form of presentation [26,27]. However, as the majority of patients receiving anti-TNF for inflammatory diseases other than psoriasis do not develop paradoxical psoriasis, some authors postulate that there must be an individual genetic background or unknown environmental factors that could predispose those patients since both the development and the latency period of paradoxical psoriasis is very variable [26,27,28,29,30]. In this regard, skin lesions derived from induced psoriasis present more plasmacytoid dendritic cells (pDCs), fewer T-cells, and selective overexpression of type I interferons, compared to classical psoriasis [31]. Anti-TNF drugs inhibit pDCs maturation, thus prolonging type I interferon expression. As a result, increased levels of type I interferon trigger the skin phenotype of paradoxical psoriasis in a T-cell-independent manner [32].

Thus, the aim of this study was to identify CNVs associated with the different clinical types of psoriasis that could explain the clinical-therapeutic differences found between them in clinical practice. As a secondary objective, we intended to assess the potential differences in genetic background between anti-TNF-induced PPP and the different clinical types of non-induced psoriasis, including psoriatic PPP.

## 2. Materials and Methods

### 2.1. Study Design and Population

A longitudinal prospective observational study was conducted with 5 mm punch biopsy samples from skin with psoriasis. The samples were collected before treatment initiation according to the washing periods established in the exclusion criteria. Fresh tissue samples were cryopreserved in liquid nitrogen immediately after being obtained.

To study the primary objective, the samples from 4 different groups of patients, all of them older than 18 years, were included according to the following inclusion criteria (Figure 1A):-Group 1. Moderate–severe psoriasis Vulgaris: with PASI > 10 and/or BSA > 10%.-Group 2. Predominant scalp psoriasis. The patients with a scalp involvement greater than 50%. The presence of psoriasis plaques outside the scalp is restricted up to a maximum of 5% of the body surface.-Group 3. Hyperkeratotic palmoplantar psoriasis. The patients with plaque psoriasis and predominant hyperkeratotic palmar and/or plantar lesions. The presence of psoriasis plaques outside the palms and soles is restricted up to a maximum of 5% of the body surface.-Group 4. PPP. The patients with persistent (>3 months) sterile macroscopically visible pustular palmar and/or plantar lesions, with or without psoriasis Vulgaris.

The samples were analyzed in a methylation microarray. Thus, the patients were subjected to strict exclusion criteria to avoid potential bias that could affect DNA methylation. Likewise, the patients were excluded according to the following exclusion criteria: (1) pregnant or lactating women, (2) patients with malignant disease diagnosed in the previous 5 years except for non-melanoma skin cancer, (3) patients infected with human immunodeficiency virus (HIV) hepatitis B virus (HBV) and hepatitis C virus (HCV), or suffering from serious active infectious diseases, (4) patients treated with non-biological systemic treatments (methotrexate, retinoids, cyclosporine or phototherapy) 1 month before sample collection, (5) patients treated with biological drugs in the 3 months before sample collection, (6) patients treated with topical steroids in the 2 weeks before sample collection.

To study the secondary objective, palmoplantar skin biopsies were taken from 6 adult patients with inflammatory diseases other than psoriasis who developed de novo PPP while being treated with TNF-blocking drugs (Figure 1A).

The protocol and informed consent followed the principles outlined in the Declaration of Helsinki for all human experimental investigations and complied with Spanish legislation on biomedical and clinical research. The protocol and informed consent were approved by the Ethics Committee for Clinical Research of Hospital Universitario de la Fe (Valencia, Spain). Henceforth, sample processing and analysis were the same for the study of both objectives. Informed consent was obtained from the participants involved.

### 2.2. Genotyping

DNA was extracted from the skin samples using an Omega D3396-02 extraction kit (Bio-Tek system). The DNA integrity number was obtained with a 2100 BioAnalyzer (Agilent). Later, an EZ DNA Methylation Kit (Zymo Research) was used for the bisulphite conversion of 1000 ng of genomic DNA. Genome-wide DNA methylation analysis was performed with a high-density array Illumina Infinium Human Methylation (EPIC) on the skin samples according to the manufacturer’s protocol.

### 2.3. CNV Analysis

The CNVs were extracted from the methylation arrays and analyzed as described previously [25]. Briefly, the IDAT data were pre-processed using the R minfi package [33]. Next, conumee [34] was used to merge the intensity values of the unmethylated and methylated probes of each of the CpG sites. The intensity values were normalized with a series of controls obtained from the minfi.Data.EPIC package. Then, the DNA was fragmented into bins, which are DNA regions comprising 15 neighboring CpGs whose size ranges from 50,000 to 150,000 base pairs and whose number is constant for each patient (16,272) (Figure 1B).

### 2.4. Statistical Analysis

Significant CNVs were obtained by comparing the intensity values of each bin between the different groups analyzed using the Student´s *t*-test. Bonferroni correction was applied. Only the Bonferroni-adjusted *p*-values lower than 0.05 were considered significant.

## 3. Results

### 3.1. Study Subjects

This study included 39 patients with different patterns of psoriasis and lesions from six patients with anti-TNF-induced PPP (Table 1). The patients with idiopathic psoriasis (mean age ± SD, 56.15 ± 13.76 years) were slightly older than patients with anti-TNF-induced PPP (49.17 ± 11.11 years) (Table 1).

### 3.2. CNVs Associated with the Different Clinical Phenotypes of Psoriasis

Initially, we wanted to determine whether there was a genetic predisposition within the different subtypes of plaque psoriasis that could explain the clinical differences and the variability in the therapeutic response. For that purpose, we carried out the following CNV comparisons of the skin samples: scalp (N = 8) vs. non-scalp plaques (classical) (N = 8); scalp (N = 8) vs. hyperkeratotic palmoplantar (N = 12). We also analyzed scalp and non-scalp (N = 16) vs. hyperkeratotic palmoplantar psoriasis (N = 12) (Figure 1A). We did not find significant differences in any of the comparisons performed.

Later, we analyzed the differences between plaque psoriasis independently of its location (scalp, non-scalp plaques, and hyperkeratotic palmoplantar; N = 28) and PPP (N = 11). No significant differences were found.

Finally, we analyzed the differences between PPP and every plaque psoriasis subtype separately. We performed the following comparisons: scalp and non-scalp plaques (N = 16) vs. PPP (N = 11), non-scalp plaques (N = 8) vs. PPP (N = 11), scalp plaques (N = 8) vs. PPP (N = 11), and hyperkeratotic palmoplantar (N = 12) vs. PPP (N = 11) (Figure 1A); we found no significant differences.

### 3.3. CNV Differences between Paradoxical Anti-TNF-Induced PPP and Different Clinical Phenotypes of Psoriasis

We also carried out a sub-analysis with the data of skin biopsies from six patients with immune-mediated diseases (uveitis [N = 1], Crohn’s disease [CD, N = 3], and rheumatoid arthritis [RA, N = 2]) who had developed a PPP induced by anti-TNF drugs (2 infliximab, 1 adalimumab, 1 etanercept, 1 certolizumab pegol, 1 golimumab).

We compared the CNVs derived from the different clinical phenotypes of psoriasis (N = 39) with anti-TNF-induced PPP (N = 6) (Table 2, Figure 1A). We found two significant bins harboring five different genes: *LOC100129540*, *MIR599*, *MIR875 RASGRF1*, and *VPS13B* (Table 3, Appendix A).

In addition, we tried to find differences in CNVs between the patients suffering from idiopathic PPP (N = 11) and the patients with anti-TNF-induced PPP (N = 6) (Figure 1A). We did not find any significant difference between both groups.

## 4. Discussion

In this study, we analyzed CNVs associated with the different phenotypes (plaques vs. pustules) and predominant locations of psoriasis (scalp, non-scalp, and palmoplantar plaques) as well as with anti-TNF-induced PPP.

CNVs are known to be involved in psoriasis pathogenesis and the response to biologics treatments; however, few studies have studied characteristic CNVs of each type of psoriasis because most of the studies focused on classic trunk and limbs plaque psoriasis [18,19,20,21,22]. To our knowledge, this is the first study that analyzes CNVs associated with different types of psoriasis (including special localizations such as scalp and palmoplantar psoriasis) through the direct study of psoriatic tissue. A previous study genotyping IL36RN, CARD14, and AP1S3 failed to find CNVs associated with PPP in those genes (N = 258) [9]. By these results, we did not find any CNV differentially associated with PPP skin lesions concerning other psoriasis phenotypes. Neither did we find CNV differences between the different locations of psoriasis. Therefore, other genetic factors or non-genetic differences, such as epigenetic changes or even protein modifications, could explain the etiopathogenesis of the different clinical phenotypes found in psoriasis.

We determined that the studied phenotypic variations of psoriasis are not related to underlying CNV changes, suggesting that other types of non-genetic regulations may be involved and should be carefully studied. To detect CNVs associated with each type of psoriasis, further analysis should be performed including the skin samples of healthy volunteers.

Additionally, we searched for differences in skin CNVs between anti-TNF-induced PPP and different non-induced clinical types of psoriasis, including idiopathic PPP. We found two CNVs that harbor five genes, LOC100129540, MIR599, MIR875 RASGRF1, and VPS13B, which participate in diverse functions such as lipids’ transfer or the regulation of gene expression. For instance, GTPase RASGRF1 participates in inflammatory diseases such as RA [35]. Therefore, the difference found in RASGRF1 CNV between anti-TNF-induced PPP and non-induced psoriasis could be related to the fact that two of the patients developing induced psoriasiform reactions presented RA. So, further studies involving a higher number of patients are required in order to analyze the influence of RA in the development of psoriasiform reactions.

A previous study from our laboratory found five SNPs (rs11209026 in IL23R, rs10782001 in FBXL19, rs3087243 in CTLA4, rs651630 in SLC12A8, and rs1800453 in TAP1) associated with anti-TNF-induced paradoxical psoriasis reactions [36]. None of the CNVs described in the present study includes these genes. These results may be partially explained by the fact that CNVs and SNPs are complementary but different factors contributing to the appearance of psoriasiform reactions. Moreover, the aforementioned study focused on biomarkers of anti-TNF-induced psoriasiform reactions in patients suffering from existing psoriasis. In addition, when we directly compared anti-TNF-induced PPP with idiopathic PPP, we found no significant differences in CNVs. Thus, the genetic background of the patients with non-induced psoriasis could also help to explain these discrepancies.

Herein, we have followed a statistical strategy that allows CNVs quantification from methylation arrays using bioinformatics tools based on a previous publication from our group [25]. This strategy allows for maximizing the information gathered with such an expensive technique as methylation microarrays. In addition, CNV determination from methylation arrays provides similar results to the gold-standard technique for CNV analysis (CGH arrays) [37,38]. Nevertheless, quantifying CNV differences between two groups is still very challenging since most of these tools are focused on CNV representation [37,39,40]. Although different R packages allow CNV determination from methylation data, some of them are still under development (EpiCopy, cnAnalysis 450K) or outdated (CopyNumber450K). Meffil and RnBeads were not applied since they provide individual CNV information for each CpG but do not integrate these differences into CNV regions. Other R packages such as conumee and ChAMP allow the determination of segments of CNVs affecting long DNA regions (100,000 and 6,000,000 base pairs). However, CNVs affecting long DNA regions could cause more severe pathologies, such as some types of cancer, such as glioblastoma [39], and are less likely to be involved in psoriasis. Moreover, the segments generated by ChAMP and conumee present different sizes, which complicate the comparisons of the results. Thus, based on our previous experience [32], we focused on Bin analysis using conumee. Bins are regions that contain 15 neighboring CpGs, and their number is constant for each patient (16,272). However, these results should be validated with alternative techniques in order to determine if they are method-dependent. Another limitation is the unclear relationship between the intensity values and the number of CNV copies. However, certain publications consider a CNV gain when the log2 ratio is higher than 0.2 and a loss if it is lower than 0.2 [41,42,43].

The number of patients in our cohort is limited, but it must be taken into account that we included infrequent subtypes of psoriasis (i.e., palmoplantar pustulosis (PPP) has a prevalence of 0.01–0.05% [44] and hyperkeratotic palmoplantar psoriasis affects approximately 12% of all patients with psoriasis [5]). For this kind of study, power analysis is not straightforward, and the models used for sample size determination are not easily applicable to CNV analysis [45]. However, if we consider a linear association between the phenotype and the variation and a 0.5 effect size [46], for an alpha of 0.05 and a power of 75%, we would require 44 patients. Additionally, this study was based on a methylation array, which is one of its strengths, and the sample size is larger than that used in other methylation studies (N = 12 patients) [47], (N = 24 patients) [48]. On the other hand, the different genetic backgrounds between the patients with psoriasis and the patients with different inflammatory diseases, including those who developed anti-TNF-induced PPP, means that the differences found between their CVNs should be interpreted with caution and confirmed with larger studies.

## 5. Conclusions

We found no differences in CNVs between different psoriasis phenotypes. The absence of genetic mechanisms to explain the marked clinical differences and therapeutic response to the same treatments found between these subgroups in clinical practice suggests that there must be epigenetic mechanisms (DNA methylation, microRNAs, histone modifications) involved in such differences. The study of epigenetics in the different clinical forms of psoriasis could provide a more comprehensive understanding of this disease, potentially leading to new therapeutic targets for each of the clinical forms. In summary, these results may help to predict which patients could develop psoriasiform reactions. Nevertheless, the clinical relevance of these results should be confirmed using larger cohorts and different techniques.

## Figures and Tables

**Figure 1 jpm-12-01452-f001:**
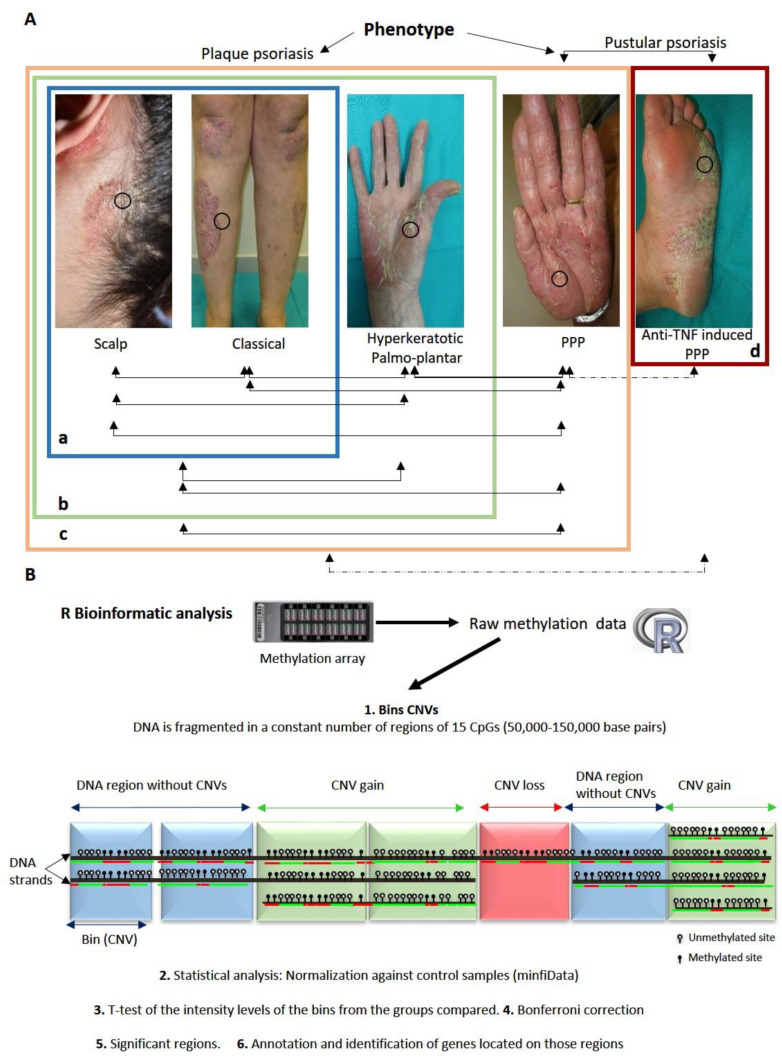
Workflow of the analysis performed in this study. (**A**) Summary of the different comparisons that were performed. Squares show the different subgroups that were used for the analysis: Blue (a) Scalp and non-scalp psoriasis; Green (b) Scalp, non-scalp, and hyperkeratotic palmoplantar plaques; Orange (c) Idiopathic psoriasis including scalp, non-scalp psoriasis, hyperkeratotic palmoplantar plaques, and PPP; Red (d) Anti-TNF-induced psoriasiform PPP. Circles show L skin. Arrows represent comparisons performed between the different types of psoriasis. Dashed arrows represent the comparison between lesional skin of idiopathic psoriasis and anti-TNF-induced psoriasiform PPP. (**B**) Bioinformatic pipeline used to obtain CNVs from raw methylation data. It comprises different stages: (1) Bins extraction from methylation raw data with the R package conumee; (2) Statistical analysis: Normalization against control samples (minfiData); (3) T-test of the intensity levels of the bins from the groups compared; (4) Bonferroni correction; (5) Selection of the significant CNVs; (6) Annotation and identification of genes located on CNVs. Abbreviation: L: lesional; PPP: Palmoplantar pustulosis.

**Table 1 jpm-12-01452-t001:** Data of patients’ demographic and anamnestic characteristics.

		Psoriasis (N = 39)	Anti-TNF-Induced PPP (N = 6)
Age (years)		56.15 ± 13.76	49.17 ± 11.11
Females		24 (61.5%)	3 (50%)
Height (m)		1.65 ± 0.08	1.65 ± 0.07
Weight (Kg)		80.41 ± 14.70	71.83 ± 9.47
Body mass index (Kg/m^2^)		29.54 ± 5.02	26.67 ± 5.50
Smoker		31 (79.5)	5 (83.3)
Time of evolution of psoriasis (years)		12.54 ± 15.97	
Diabetes (%)		9 (23.1)	0 (0.0)
Metabolic syndrome (%)		21 (53.8)	1 (16.7)
Arterial hypertension (%)		21 (53.8)	3 (50.0)
Dyslipidemia (%)		27 (69.2)	2 (33.3)
Psoriatic arthritis (%)		11 (28.2)	3 (50.0)
Hyperthyroidism		7 (17.9)	0 (0.0)
Hepatic steatosis		16 (41.0)	0 (0.0)
Ever smokers (%)		31 (79.5)	5 (83.3)
Subset of psoriasis	Non-scalp (%)	8 (20.5%)	
	Scalp (%)	8 (20.5%)	
	Hyperkeratotic palmoplantar (%)	12 (30.8%)	
	PPP (%)	11 (28.2%)	6 (100%)
Induced psoriasiform reactions	Anti-TNF-induced PPP latency (months) (%)		34.33 ± 36.30
	Inflammatory Bowel disease (%)		3 (50.0)
	Rheumatoid arthritis (%)		2 (33.3)
	Uveitis (%)		1 (16.7)

Abbreviations: PPP: palmoplantar pustulosis. TNF: Tumor necrosis factor. Data are shown either as mean ± standard deviation (SD), or as number and percentage (%).

**Table 2 jpm-12-01452-t002:** Summary of significant CNVs associated with anti-TNF-induced psoriasiform reactions.

Comparison	Chromosome Location	Adj. *p*	Log2 Ratio #	Length (Base Pairs)	Genes
**Idiopathic psoriasis** (N = 39) vs. **induced psoriasiform reactions** (N = 6)	chr8:100000001-100750000	0.045	−0.165	750,000	*VPS13B*, *MIR599*, *MIR875*
	chr15:79250001-79350000	0.044	0.177	100,000	*RASGRF1*, *LOC100129540*

Abbreviations. PPP: Palmoplantar pustulosis. Adj. P: *p* value adjusted by Bonferroni correction. Log2 ratio #: patients with psoriasiform reactions are referred to patients without psoriasiform reactions. Length: length of CNVs regions.

**Table 3 jpm-12-01452-t003:** Summary of the genes located on the significant bins obtained and their function.

Comparison	Genes ID	Gene Name	Function
**Idiopathic psoriasis** (N = 39) vs. **induced psoriasiform reactions** (N = 6)	*LOC100129540*	Uncharacterized LOC100129540	Related to coronary heart disease and bipolar disorder
	*MIR599*	MicroRNA 599	Post-transcriptional regulation of gene expression
	*MIR875*	MicroRNA 875	Post-transcriptional regulation of gene expression
	*RASGRF1*	Ras Protein Specific Guanine Nucleotide Releasing Factor 1	Promotes the exchange of Ras-bound GDP by GTP
	*VPS13*	Vacuolar Protein Sorting 13 Homolog A	Allows the formation or stabilization of ER-mitochondria contact sites which facilitates lipids’ transfer between the ER and mitochondria

Abbreviations: PPP: Palmoplantar pustulosis.

## Data Availability

The datasets use and/or analyzed during the current study are available from the corresponding author upon reasonable request.

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
