# Peer review of "CNVs Associated with Different Clinical Phenotypes of Psoriasis and Anti-TNF-Induced Palmoplantar Pustulosis"

_jpm, 2022, doi:10.3390/jpm12091452_

Round 1
Reviewer 1 Report
This is a very novel article in the field of psoriasis. It is excellent work. The methodology is perfect and there are few comments to add. In my opinion, the English style should be reviewed. Table S1 is missing (it seems to be the one that appears as Table 1).
Author Response
Thank you very much for your supportive comments. We have corrected the English style.
There is no supplementary table 1. We apologize for this mistake, we have deleted the legend of this table and corrected it in the manuscript.

Reviewer 2 Report
The authors investigated CNVs in different types of psoriasis and detected no differences. The followings are some concerns.
1) The background of investigating CNVs is not fully explained, and readers may feel some difficulty in understanding the aim of this study. The authors could more concretely explain on CNVs, such as which kind of CNVs have been elucidated so far, and the effect which CNVs may have on psoriasis pathogenesis.
2) The study was negative result. However, if the background of this study were fully explained, it might be of interest to the readers.
3) Is the population size enough to detect differences in this study. Because it is a surprise not to detect any differences between ordinary PPP and TNF-induced PPP, who may have different inflammatory diseases background such as rheumatoid arthritis or IBD.
4) Has it been already reported if there are any CNVs differences between patients with RA or IBD without developing paradoxical reaction and RA or IBD patients who developed paradoxical reaction?
Author Response
Thank you very much for your comments and suggestions for improvement.
1) The background of investigating CNVs is not fully explained, and readers may feel some difficulty in understanding the aim of this study. The authors could more concretely explain on CNVs, such as which kind of CNVs have been elucidated so far, and the effect which CNVs may have on psoriasis pathogenesis.
Answer: Thank you very much for this comment. According to the suggestion of the reviewer 2, we have improved the introduction, reinforcing the role of CNVs in the pathogenesis of psoriasis as well as in its possible predictor of response to treatment, based on previous publications.
2) The study was negative result. However, if the background of this study were fully explained, it might be of interest to the readers.
Answer: As we have mentioned previously, we have modified the introduction. We have also emphasized the relevance of our study because it has been performed directly in the skin tissue affected by the disease, and not in peripheral blood as in previous publications. Most patients suffering from psoriasis do not show a generalized disease whereas the cutaneous involvement is focused on specific areas. This, this fact suggests that studying the tissue that clinically develops the disease could provide more reliable data on the pathophysiology of psoriasis.
3) Is the population size enough to detect differences in this study. Because it is a surprise not to detect any differences between ordinary PPP and TNF-induced PPP, who may have different inflammatory diseases background such as rheumatoid arthritis or IBD
Answer: As reflected in the discussion, for this kind of studies power analysis is not straightforward, the models used for sample size determination are not easily applicable to CNV analysis [ref 46]. On the other hand, certain phenotypes of psoriasis, such as PPP, are of very low prevalence (0.01–0.05% [ref. 44]), which makes it very difficult to obtain samples in these patients especially considering that our study was performed directly on affected skin and not on peripheral blood. The fact that no differences were found despite the different background of RA or IBD could be due to the low sample, but also to the fact that there are no CNVs involved in a relevant way in their pathophysiopathogenesis. Alternatively, the absence of significant CNVs between ordinary and induced PPP sugggest that there must be epigenetic mechanisms (DNA methylation, microRNAs, histone modifications) involved in such differences.
4) Has it been already reported if there are any CNVs differences between patients with RA or IBD without developing paradoxical reaction and RA or IBD patients who developed paradoxical reaction?
Answer: We have not found any publication on the subject, probably due to the difficulty of obtaining good casuistry due to the low prevalence of these clinical pictures, as well as the added difficulty of being able to perform a correct sampling of the affected skin tissue at the acute stage of the outbreak. All this would give more value to our results, despite the lack of a large sample size as suggested above.

Round 2
Reviewer 2 Report
The authors revised the manuscript well.
Minor comments.
1) psoriasis Vulgaris should be psoriasis vulgaris.
2) There are some other grammatical errors in the text, which should be amended.
Author Response
Thank you very much for your comments. We have reviewed the entire manuscript and corrected the grammatical and typographical errors. We have submitted a new version of the manuscript with the modifications highlighted and a clean version.